# Fluorescence-Based Analysis of Noncanonical Functions of Aminoacyl-tRNA Synthetase-Interacting Multifunctional Proteins (AIMPs) in Peripheral Nerves

**DOI:** 10.3390/ma12071064

**Published:** 2019-04-01

**Authors:** Muwoong Kim, Hyosun Kim, Dokyoung Kim, Chan Park, Youngbuhm Huh, Junyang Jung, Hyung-Joo Chung, Na Young Jeong

**Affiliations:** 1Department of Anatomy and Neurobiology, College of Medicine, Kyung Hee University, 26, Kyungheedae-ro, Dongdaemun-gu, Seoul 02447, Korea; qltnsdl9@khu.ac.kr (M.K.); hoysun0407@khu.ac.kr (H.K.); dkim@khu.ac.kr (D.K.); psychan@khu.ac.kr (C.P.); ybhuh@khu.ac.kr (Y.H.); 2Department of Biomedical Science, Medical Research Center for Bioreaction to Reactive Oxygen Species and Biomedical Science Institute, Graduation School, Kyung Hee University, 26, Kyungheedae-ro, Dongdaemun-gu, Seoul 02447, Korea; 3Department of Anesthesiology and Pain Medicine, College of Medicine, Kosin University, 262, Gamcheon-ro, Seo-gu, Busan 49267, Korea; 4Department of Anatomy and Cell Biology, College of Medicine, Dong-A University, 32, Daesingongwon-ro, Seo-gu, Busan 49201, Korea

**Keywords:** aminoacyl-tRNA synthetase-interacting multifunctional proteins (AIMPs), Schwann cells, Wallerian degeneration, noncanonical functions, fluorescence-based analysis

## Abstract

Aminoacyl-tRNA synthetase-interacting multifunctional proteins (AIMPs) are auxiliary factors involved in protein synthesis related to aminoacyl-tRNA synthetases (ARSs). AIMPs, which are well known as nonenzymatic factors, include AIMP1/p43, AIMP2/p38, and AIMP3/p18. The canonical functions of AIMPs include not only protein synthesis via multisynthetase complexes but also maintenance of the structural stability of these complexes. Several recent studies have demonstrated nontypical (noncanonical) functions of AIMPs, such as roles in apoptosis, inflammatory processes, DNA repair, and so on. However, these noncanonical functions of AIMPs have not been studied in peripheral nerves related to motor and sensory functions. Peripheral nerves include two types of structures: peripheral axons and Schwann cells. The myelin sheath formed by Schwann cells produces saltatory conduction, and these rapid electrical signals control motor and sensory functioning in the service of survival in mammals. Schwann cells play roles not only in myelin sheath formation but also as modulators of nerve degeneration and regeneration. Therefore, it is important to identify the main functions of Schwann cells in peripheral nerves. Here, using immunofluorescence technique, we demonstrated that AIMPs are essential morphological indicators of peripheral nerve degeneration, and their actions are limited to peripheral nerves and not the dorsal root ganglion and the ventral horn of the spinal cord.

## 1. Introduction

Aminoacyl-tRNA synthetases (ARSs) are enzymes involved in attaching appropriate amino acids to the 3’-termini of tRNAs in translation [1]. There are 20 types of ARSs, which exist as complexes with other ARSs or as free forms in living cells [2]. ARS-interacting multifunctional proteins (AIMPs), which are cofactors that assist ARSs in protein synthesis, include p43 (AIMP1), p38 (AIMP2), and p18 (AIMP3). In the cytoplasm of mammalian cells, AIMPs form huge multisynthetase complexes with nine ARSs (aspartyl-tRNA synthetase, glutamyl-tRNA synthetase, prolyl-tRNA synthetase, isoleucyl-tRNA synthetase, lysyl-tRNA synthetase, leucyl-tRNA synthetase, methionyl-tRNA synthetase, glutaminyl-tRNA synthetase, and arginyl-tRNA synthetase) and play important roles in protein synthesis [3,4]. AIMPs are typically closely associated with ARSs and provide structural stability to their complexes and assist in the enzymatic activation of ARSs for protein synthesis [2,4,5]. Several recent studies have reported nontypical (noncanonical) functions of AIMPs in living organisms. For example, AIMP1 was shown to affect the cell signaling associated with inflammation, angiogenesis, apoptosis, and wound healing [6,7,8], and AIMP2 and AIMP3 were shown to be related to the regulation of the cell cycle and tumor suppression [9,10]. However, these noncanonical functions of AIMPs have not been studied in peripheral nerves, which are essential structures for motor and sensory behaviors in mammals.

Schwann cells are components of the peripheral nerves with functions related to peripheral nerve degeneration and regeneration as well as saltatory conduction for fast signal transduction from central to peripheral organs. Therefore, dysfunctions in Schwann cell dynamics cause various problems in peripheral organs innervated by peripheral nerves. In particular, mechanical nerve injury and neurodegenerative processes related to aging and metabolic or exogenous toxicity are main causes of peripheral neurodegeneration in mammals [11,12,13,14]. However, with the exception of the physiological evaluation of peripheral neurodegenerative dysfunction, such as nerve conduction velocity, no effective methods for morphological quantification of peripheral neurodegenerative dysfunction are available in mammals.

In this study, we demonstrated that AIMPs are highly expressed in Schwann cells in injured peripheral nerves and that their levels are decreased during the process of nerve regeneration. Therefore, our results suggest that the noncanonical functions of AIMPs act as morphological intracellular indicators of peripheral nerve degeneration.

## 2. Materials and Methods

### 2.1. Animals

The experimental procedures performed on male C57BL/6 mice (Orientbio, Seongnam, Korea) were carried out in accordance with the approved guidelines from Kyung Hee University committee on animal research [KHUASP(SE)-16-043-1]. Sciatic nerves of five-week-old mice were used in our experiment. We anesthetized animals, made a small incision unilaterally to expose the sciatic nerve, and crushed the sciatic nerve with flat forceps. At 3, 7, 14, 28 days after nerve crush, mice were euthanized by CO_2_, and the sciatic nerves, dorsal root ganglion (DRG), and spinal cord were removed for further experiments. From the injured sciatic nerve, only distal segments were collected.

### 2.2. Western Blot Analysis

Extracted sciatic nerves, DRGs and ventral horn of spinal cords were homogenized in radioimmunoprecipitation assay buffer [RIPA; 25 mM Tris-HCl (pH 7.6), 150 mM NaCl, 1% Triton NP-40, 0.1% sodium dodecyl sulfate (SDS; Thermo, Waltham, MA, USA)] with protease inhibitor mixture (Roche Molecular Biochemicals, Nutley, USA) and 1 mM Na_3_VO_4_. Western blot analyses were performed following protein blotting guide (Bio-Rad, Hercules, CA USA). The following primary antibodies were used: anti-AIMP1 (Neomics, Suwon, Korea), anti-AIMP2 (Proteintech, Chicago, IL, USA), anti-AIMP3 (Neomics, Suwon, Korea), and β-actin (Sigma, St. Louis, MO, USA).

### 2.3. Tissue Preparation

Mouse tissue samples were fixed with 4% paraformaldehyde (4% PFA) in phosphate-buffered saline (PBS) overnight at 4°C. For cryosection, samples were cryoprotected in 30% sucrose in PBS for 2 days at 4°C. Dehydrated sciatic nerves and DRGs were embedded in optimal cutting temperature (OCT) compound and transverse sections (16 μm thickness) were prepared by using vibratome in cryostat (Leica, Wetzlar, Germany). To prepare teased nerve slides, the fixed sciatic nerves were teased into single nerve fibers under a stereomicroscope. The prepared tissue section and teased nerve fibers were dried and kept at −80°C until use.

### 2.4. Immunofluorescence

Prepared samples were post-fixed with 4% PFA. After one washing with PBS and another washing with PBS including 0.3% Triton X-100 (PBS-T), followed by being blocked in blocking solution (5% bovine serum albumin and 5% fetal bovine serum in PBS-T) at room temperature (RT), samples were incubated with primary antibodies diluted in blocking solution overnight at 4 °C. The following primary antibodies were used: anti-AIMP1, anti-AIMP2, anti-AIMP3, anti-NeuN (Santa Cruz Biotechnology, CA, USA), anti-Neurofilament (NF) H&M (Millipore, Bedford, MA, USA), anti-p75 neurotrophin receptor (NTR), and anti-S100β (Sigma, St. Louis, MO, USA). Samples then were washed three times with PBS and then incubated with Alexa Fluor 488- and 594-conjugated secondary antibodies (Invitrogen, Carlsbad, CA, USA) in blocking solution for 2 hours at RT, washed three times with PBS, and mounted with cover glass.

### 2.5. Polymerase Chain Reaction (PCR)

Total RNA was isolated from mouse sciatic nerves using Tri Reagent (MRC, Cincinnati, OH, USA), and first-strand cDNA was obtained with SuperScript Reverse transcriptase (Invitrogen, Carlsbad, CA, USA), according to the manufacturer’s protocol. Real-time quantitative PCR (qPCR) was performed with SYBR green master mix (Takara, Kusatsu, Japan). Primers for qPCR are: AIMP1 (FW: 5’-TTTCTCTGCCGATTCTGGGGA-3’, RV: 3’-CCTGCTGCTTGAGATATTCGAT-5’), AIMP2 (FW: 5’-CGTGCAGGAAACATCCGA-3’, RV: 3’-GTTACGTCCAAGTCTGCATCT–5’), AIMP3 (FW: 5’-GACTGAAGCCGGGGAATAAGT-3’, RV: 3’-TAGACTCGGGCCATTGTTTGT–5’) and Glyceraldehyde-3-phosphate dehydrogenase (GAPDH) (FW: 5’–TGCACCACCAACTGCTTAGC-3’, 3’-GGCATGGACTGTGGTCATGA–5’). Semi-PCR was performed using G-Taq polymerase (Cosmogenetech, Seoul, Korea). Primers for semi-PCR are: AIMP1 (FW: 5’-AAAAGGAGAAGCAGCAGTCG-3’, RV: 5’-TCTGGTGAACTGGCACACAT-3’), AIMP2 (FW: 5’-AGTTGAAGGCAGCAGTCGAT-3’, RV: 5’-GACAAGATTCTCGGGCACAT-3’), AIMP3 (FW: 5’-ATCGCCACCCATCTAGTCAA-3’, RV: 5’-GACAAAACCAGCGAGACACA-3’), GAPDH (FW: 5’-CTACATGGTCTACATGTTCCAGTATG-3’, RV: 5’-AGTTGTCATGGATGACCTTGG-3’).

### 2.6. Statistical Analysis 

Statistical analysis was performed by using SPSS 21.0 software (IBM, Chicago, IL, USA). P-values are from Student’s two-tailed test, and results were expressed as mean and standard error. *p* < 0.05 was considered as statistically significant.

## 3. Results

### 3.1. AIMPs Are Highly Expressed in Dysfunctional Peripheral Nerves and Decreased in Recovered Peripheral Nerves

We used the in vivo sciatic crush model, which induces peripheral nerve degeneration and regeneration, to induce peripheral nerve dysfunction. During sciatic nerve degeneration, motor and sensory functions were diminished because all axons were degraded and disconnected from their target organs. The functions recovered during nerve regeneration because peripheral nerves have regenerative capacity. We performed qPCR analysis using primers for AIMP1–3 to determine whether AIMPs show alterations in their mRNA expression during nerve degeneration and regeneration. During nerve degeneration (from three days to one week), AIMP1–3 showed high levels of expression compared with controls (uninjured sciatic nerve) (Figure 1a). In contrast, AIMP1–3 expression levels were markedly decreased during nerve regeneration (from two to four weeks) (Figure 1a). To visually confirm qPCR results and evaluate the specificity of the primers, we performed semi-PCR analysis, which also showed that AIMP1–3 mRNA expression levels were markedly increased during nerve degeneration and suddenly decreased during nerve regeneration (Figure 1b and Appendix A).

Next, to determine the patterns of AIMP protein expression, we performed Western blotting analysis using antibodies to AIMP1–3. Similar to the patterns of AIMP mRNA expression levels, AIMP1–3 showed elevated expression at the protein level during nerve degeneration with marked decreases during nerve regeneration (Figure 2a). Quantification of AIMP expression by Western blotting also indicated marked increases in AIMP expression during nerve degeneration (Figure 2b). 

To visualize the patterns of AIMP expression during nerve degeneration and regeneration, we performed immunostaining using the same antibodies used in Western blotting. Immunostaining also showed that AIMP1–3 was highly expressed during nerve degeneration, and the levels of expression decreased during nerve regeneration, which was similar to the results of the Western blotting analysis (Figure 3a). Quantification of the immunostaining intensities indicated that the peak timing of AIMP1–3 expression was one week after sciatic nerve crush injury (Figure 3b). Taken together, these results indicated that AIMPs are highly expressed during nerve degeneration and that their increases likely represent peripheral nerve dysfunction morphologically.

### 3.2. Increases in AIMP Expression Accrue in Schwann Cells of Dysfunctional Peripheral Nerves 

Peripheral nerves contain two types of structures: peripheral axons and Schwann cells. To identify the cell types related to AIMP expression under conditions of peripheral neurodegenerative process, we performed immunostaining for AIMPs, S100 (a Schwann cell marker), and NF (an axon marker). In the control and injured sciatic nerves (three days after crush injury), AIMP1–3 was highly expressed in the injured nerves and strongly overlapped with positive signals for S100 but not NF (Figure 4a–c). Quantitative evaluation also indicated that AIMP1–3 was coexpressed with S100 but not NF (Figure 4d,e). Aside from peripheral axons, to identify whether AIMPs affect injured neuronal cell bodies of peripheral nerves, we evaluated the patterns of AIMP1–3 expression in the DRG. Immunostaining analysis indicated that AIMP1–3 was continuously expressed along with NeuN (a marker for neuronal cell bodies) regardless of condition (i.e., control or injury) (Figure 5a). These AIMP-positive signals did not overlap with S100, a marker of satellite cells, which are glial cells in the DRG (Figure 5b). Western blotting analysis also showed identical AIMP1–3 expression in both the control and injured sides of the DRG (Figure 5c) and the ventral horn of the spinal cord (Appendix A). Quantitative data indicated that there were no differences in AIMP1–3 expression in the DRG and spinal cord between the control and injured sides (Figure 5d and Appendix A). 

Interestingly, AIMPs likely show expression patterns similar to p75NTR as a marker of trans-dedifferentiation. In immunofluorescence analysis with antibodies of AIMPs and p75NTR, signals of AIMPs overlapped with p75NTR profiles and the time of expression of AIMPs coincided with that of p75NTR (Figure 6a,b). Thus, AIMPs may be used as an effective marker for trans-dedifferentiation. Taken together, these results indicate that Schwann cells highly express AIMP1–3 during peripheral nerve degeneration. Additionally, the AIMP1–3 expression patterns were not altered regardless of the position of the peripheral neurons (axon and cell body).

## 4. Discussion

### 4.1. AIMPs Act as Morphological Indicators of Dysfunctional Peripheral Nerves in Vivo

There are two types of peripheral nerves: cranial nerves projecting from the brain and spinal nerves projecting from the spinal cord. Peripheral nerves are organized much like power cables that distribute electricity from power transmission plants to appliances in the home, such as refrigerators and washing machines. That is, all organs in the body receive electrical signals from the brain via peripheral nerves to regulate their functions. As conditions that lead to peripheral nerve dysfunction are responsible for many severe disorders in mammals, including chronic pain, quadriplegia, numbness, dysphasia, and so on [15,16,17], it is important to identify morphological indicators capable of reflecting the functional condition of peripheral nerves. However, no such effective morphological indicators have been reported to date.

The results of the present study suggest that AIMPs are effective indicators for identifying dysfunctional peripheral nerves morphologically. We used the sciatic nerve crush model to investigate the functional state of peripheral nerves morphologically. After nerve crush injury, axons and myelin undergo degradation and disappear, resulting in disconnection between the central nervous system and peripheral organs, and this nerve degeneration represents peripheral nerve dysfunction. According to this model, after peripheral nerve degeneration, axons and myelin are regenerated, and their connections are recovered; this nerve regeneration represents the functional state of peripheral nerves. That is, without requiring complicated physiological measurements, the functional state of peripheral nerves can be evaluated by morphological assessments alone. However, an effective indicator for detecting dysfunctional peripheral nerves in *Mammalia* is still absolutely needed. In the present study, AIMPs were more highly expressed in injured sciatic nerves than in control (no injury) during peripheral nerve degeneration (dysfunctional state, samples from three days and one week) (Figure 1, Figure 2 and Figure 3 and Appendix A). During nerve regeneration, the disconnected nerves recovered and the levels of AIMP expression were decreased (Figure 1, Figure 2 and Figure 3 and Appendix A). These AIMP expression patterns may represent the normal and abnormal conditions of peripheral nerves, respectively.

These observations raise questions regarding those noncanonical functions of AIMPs in dysfunctional peripheral nerves that may serve as good indicators. First, AIMP1 stimulates macrophages to induce proinflammatory cytokines via p38 mitogen-activated protein kinases (p38 MAPK), extracellular signal-regulated kinases (ERK), and nuclear factor kappa B (NF-кB) cell signaling pathways [6,7]. During peripheral nerve degeneration, the degraded myelin sheath debris is removed by macrophages and Schwann cells [13]. Thus, increased AIMP1 in injured peripheral nerves may activate macrophages, which would then engulf myelin debris. Second, TGF-β-induced AIMP2 translocates into the nucleus and then binds to far-upstream element-binding protein (FUSE-binding protein, FBP) to facilitate proteolysis through ubiquitination [18]. AIMP3 is also related to the ubiquitination of lamin A via Siah1, E3 ubiquitin ligase [19]. In Schwann cells, the ubiquitin–proteasome system is involved in myelin degradation during peripheral nerve degeneration [20,21]. Thus, ubiquitination-related AIMP2 and AIMP3 may activate the ubiquitin–proteasome system and affect the process of myelin debris clearance in Schwann cells during demyelination. Taken together, the alterations of AIMP expression through their noncanonical actions may act as morphological indicators of peripheral nerve dysfunction.

### 4.2. AIMPs Indicate the Trans-Dedifferentiation of in Vivo Schwann Cells in Dysfunctional Peripheral Nerves

During peripheral nerve degeneration, Schwann cells alter their characteristics to become suitable for nerve regeneration, which is a state of functional recovery. For example, Schwann cells suppress the production of myelin-related protein [22] and enhance the expression of transcription factors to activate myelin degradation, such as c-jun, Sox-2, and notch [23], during nerve degeneration. Additionally, injured Schwann cells activate adherent molecules, neurotrophic factor receptors, and the MAPK signaling pathway [23]. Therefore, the molecularly, morphologically, and functionally altered Schwann cells during peripheral nerve degeneration are called trans-dedifferentiated Schwann cells [24]. Therefore, it is possible that the trans-dedifferentiation of Schwann cells is representative of dysfunctional conditions of peripheral nerves.

The present study indicated that in dysfunctional peripheral nerves, alteration of AIMPs expression occurred in Schwann cells but not in peripheral axons or the neuronal cell body (Figure 4 and Figure 5 and Appendix A). These results indicated that AIMPs could show their noncanonical functions only in Schwann cells but not peripheral neurons. In degenerating peripheral neurons, AIMPs likely show only the canonical functions for protein synthesis (Figure 5 and Appendix A). If peripheral neurons have noncanonical functions, AIMP1–3 would exhibit higher expression in the injured neuronal cell bodies than in normal neuronal cell bodies (Figure 5a,b). AIMPs may represent the trans-dedifferentiation of Schwann cells, as AIMPs show Schwann-cell-specific activation in injured peripheral nerves (Figure 4), and the period of AIMP expression coincides with that of Schwann cell trans-dedifferentiation during nerve degeneration and regeneration (Figure 6) [13,25,26]. Additionally, AIMP1 is known to activate p38 MAPK and ERK signaling pathways as well as proinflammatory signals [6,7]. As p38 MAPK, ERK, and c-jun activation are the main phenotypes of Schwann cell trans-dedifferentiation [27,28,29], activation of the MAPK pathway through AIMP1 is likely one of the noncanonical functions of AIMPs for the trans-dedifferentiation of Schwann cells. Taken together, these observations suggest that functions of AIMPs in dysfunctional peripheral nerves may be involved in trans-dedifferentiation of Schwann cells. However, further evaluation is needed to strengthen the noncanonical functions of AIMPs on the trans-dedifferentiation of Schwann cells.

## 5. Conclusions

AIMPs have several noncanonical functions in addition to their typical roles in protein synthesis. Their noncanonical functions strongly affect dysfunctional peripheral nerves, especially trans-dedifferentiating Schwann cells. For example, peripheral degenerative disorders in diabetic neuropathy, Charcot–Marie–Tooth disease, and Guillain–Barré syndrome in humans cause dysesthesia, speech impairment, vision changes, erectile dysfunction, and urinary incontinence via peripheral nerve dysfunction [12,30,31,32,33]. Therefore, it is important to investigate the roles of AIMPs as morphological indicators of dysfunctional peripheral nerves to study the molecular physiology of animal behavior or to diagnose human peripheral neurodegenerative diseases in their early stages.

## Figures and Tables

**Figure 1 materials-12-01064-f001:**
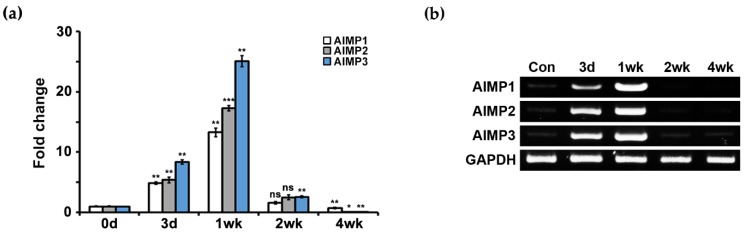
mRNA expression patterns of aminoacyl-tRNA synthetase-interacting multifunctional proteins (AIMPs) under normal and abnormal conditions of peripheral nerves. (**a**) Real-time quantitative PCR (qPCR) results showed mRNA expressions of AIMPs in sciatic nerves. Relative mRNA expressions of AIMPs were calculated by fold change; 1 was expression level of control sample (Con, not injured nerves). Fold changes of postinjury samples (3 days, 1 week, 2 weeks, 4 weeks) were derived by mRNA expression level of each sample divided by that of Con (*n* = 3, * *p* < 0.05, ** *p* < 0.01, *** *p* < 0.001, ns; *p* > 0.05). Glyceraldehyde-3-phosphate dehydrogenase (GAPDH) was used as loading control. (**b**) Semi-PCR results showed mRNA expressions of AIMPs in sciatic nerve.

**Figure 2 materials-12-01064-f002:**
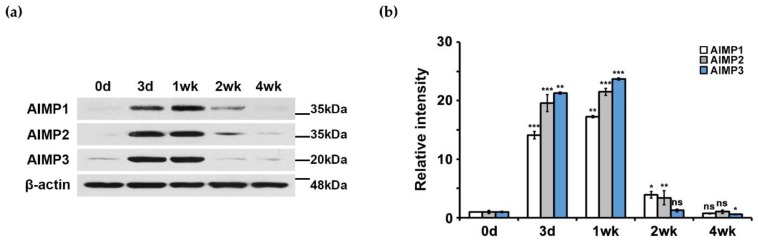
Protein expression patterns of AIMPs under normal and abnormal conditions of peripheral nerves. (**a**) Western blot analysis showed protein expression patterns of AIMPs in sciatic nerves. (**b**) Relative intensities of protein expressions of AIMPs were calculated by fold change; 1 was expression level of Con. Fold changes of postinjury samples (3 days, 1 week, 2 weeks, 4 weeks) were derived by protein expression level of each sample divided by that of Con (*n* = 3, * *p* < 0.05, ** *p* < 0.01, *** *p* < 0.001, ns; *p* > 0.05). β-actin was used as loading control.

**Figure 3 materials-12-01064-f003:**
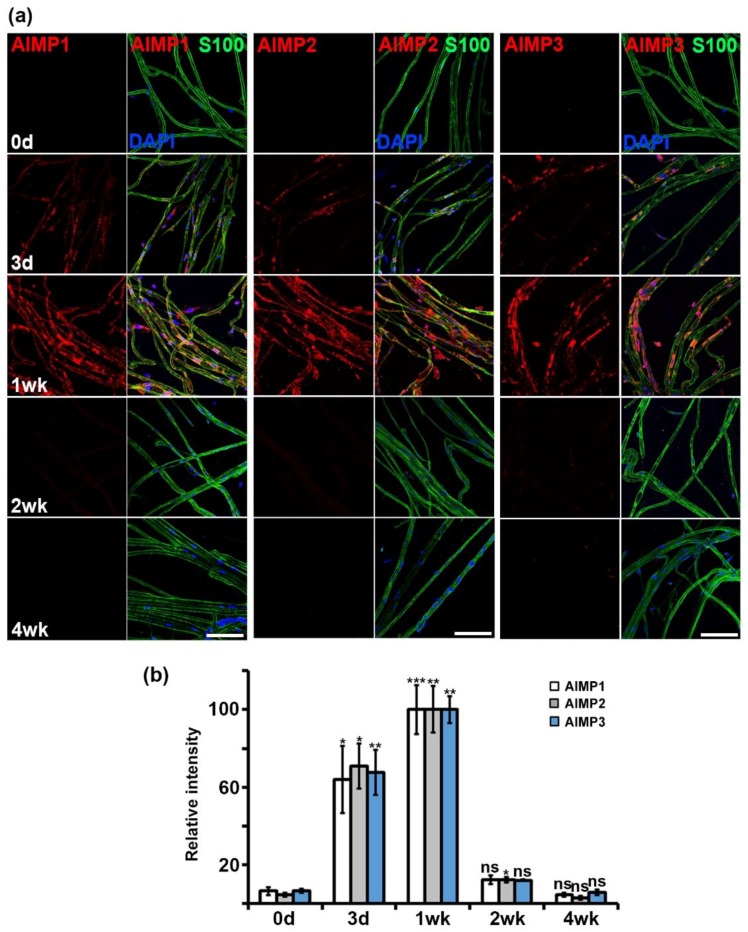
Morphological expression patterns of AIMPs under normal and abnormal conditions of peripheral nerves. (**a**) Immunostaining showed colocalization of AIMPs (red) with S100 (green, Schwann cell marker) in teased sciatic nerves. Expressions of AIMPs were maximized at 1 week and gradually decreased from 1 week after injury. Scale bar = 100 μm. (**b**) Relative intensities of AIMPs were calculated by strength of each AIMP signal; relative intensity of 1 week was 100. Relative intensities of postinjury samples (Con, 3 days, 2 weeks, 4 weeks) were derived by strength of signal of each sample divided by that of 1 week (*n* = 3, * *p* < 0.05, ** *p* < 0.01, *** *p* < 0.001, ns; *p* > 0.05).

**Figure 4 materials-12-01064-f004:**
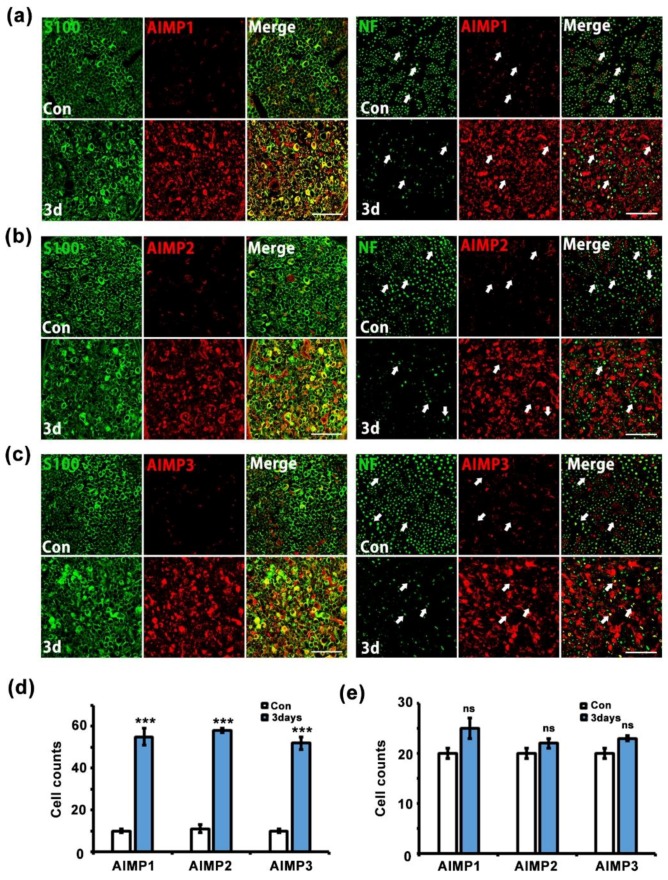
AIMPs are increased in Schwann cells under dysfunctional condition of peripheral nerves. (**a**–**c**) Immunostaining showed colocalization of AIMP1-3 (red) with S100 (green), not neurofilament (NF; green, axonal marker) after injury. Scale bar = 50 μm. (**d**,**e**) Cell counts were derived by S100/AIMPs double-positive cells or NF/AIMPs double-positive cells at Con or 3 days postinjury (*n* = 3, * *p* < 0.05, ** *p* < 0.01, *** *p* < 0.001, ns; *p* > 0.05).

**Figure 5 materials-12-01064-f005:**
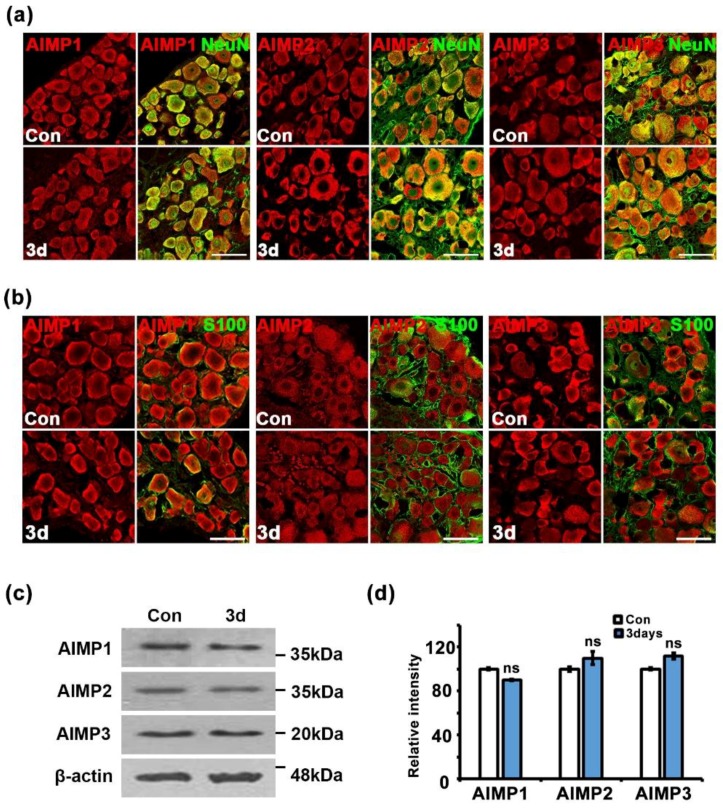
Alteration of AIMPs expression is limited to Schwann cells under dysfunctional condition of peripheral nerves. (**a**) Immunostaining showed colocalization of AIMPs (red) with NeuN (green, neuronal marker) in dorsal root ganglia (DRGs). Scale bar = 100 μm. (**b**) Immunostaining showed colocalization of AIMPs (red) with S100 (green, satellite cell marker) in DRG. Scale bar = 50 μm. (**c**) Western blot analysis showed expression patterns of AIMPs in DRGs with or without nerve injury. β-actin was used as loading control. (**d**) Relative intensities of immunostaining of AIMPs were calculated by fold change of the injured DRG compared to that of Con; Con was 100 (*n* = 3, ns; *p* > 0.05).

**Figure 6 materials-12-01064-f006:**
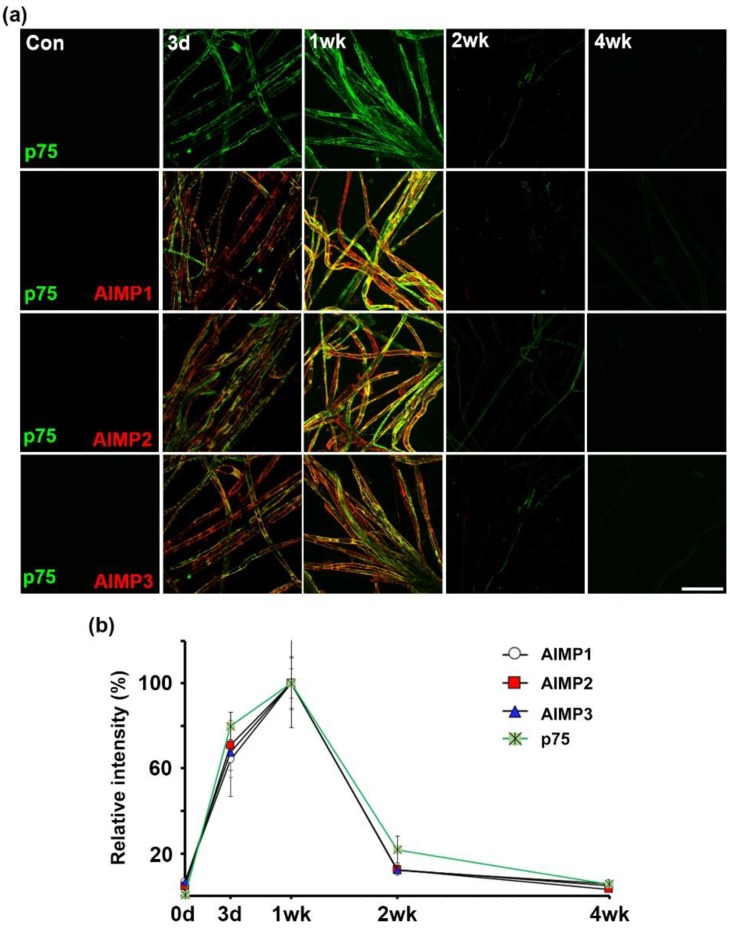
Alteration of AIMPs expressions is similar to p75 neurotrophin receptor (NTR) as a marker of Schwann cell trans-dedifferentiation. (**a**) Immunostaining showed colocalization of AIMPs (red) with p75NTR (green) in sciatic nerve fibers. Scale bar = 100 μm. (**b**) Relative intensities of immunostaining of AIMPs and p75NTR were calculated by fold change of the degenerating and regenerating nerve fibers compared to that of samples at 1 wk after nerve injury, time-course-dependently (*n* = 3, *p* < 0.001).

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
