# Peer review of "Fluorescence-Based Analysis of Noncanonical Functions of Aminoacyl-tRNA Synthetase-Interacting Multifunctional Proteins (AIMPs) in Peripheral Nerves"

_materials, 2019, doi:10.3390/ma12071064_

Round 1
Reviewer 1 Report
Aminoacyl tRNA synthetases (ARSs) and related proteins have recently gained a large attention due to their wide range of non-canonical functions that are involved in many aspects of biology. AIMPs are proteins involved in multi-tRNA synthetase complexes in mammalian cells and play a role in assisting ARS functions as well as involved in other non-canonical functions. Here in this paper, authors studied the expressions of AIMPs in peripheral nerves in terms of mRNA and protein levels and found that expression of all three AIMPs were highly upregulated in dysfunctional nerves while stayed normal in recovered nerves. Expressions were seen in Schwann cells but not in axon. Taken together, this is the first demonstration of AIMPs expressed in dysfunctional nerves and predicted to have non-canonical roles involved in for example, macrophage activation to remove myelin debris and proteasome activation. They are perhaps also important for trans-dedifferentiation of Schwann cells in dysfunctional perpheral nerves. More importantly, AIMPs can be morphological indicators of peripheral nerve degeneration. Many novel information and novel insights are included in this paper.
Some minor comments.
Page 6, in Figure 4; explanation for panel (e) is missing.
Page 7, in Figure 5; (c) should be (d) and the explanation for panel (c) is missing. Also, in (b), “co-localization of AIMPs (red) with S100 (green)” is wrong. It did not co-localize with S100 marker.
Author Response
Point 1: Page 6, in Figure 4; explanation for panel (e) is missing
Response 1: We thank you for your comment and agree with your concern regarding missing a figure indication and legend. In previous manuscript, (d) and (e) have same legends. Thus, we have changed (d) to (d,e) and the revised portion of the manuscript and new text appear on page 6, lines 177.
Point 2: in Figure 5; (c) should be (d) and the explanation for panel (c) is missing. Also, in (b), “co-localization of AIMPs (red) with S100 (green)” is wrong. It did not co-localize with S100 marker.
Response 2: We thank you for your comment and agree with your concern regarding missing legend of Figure 5c and unappropriated indications. Following your comment, we have added “Western blot analysis showed expression patterns of AIMPs in dorsal root ganglia (DRG) with or without nerve injury. β-actin was used as loading control” to legend of Figure 5c in revised manuscript. The revised portion of the manuscript and new text appear on page 7, lines 200 ~ 202.

Reviewer 2 Report
Overall a good paper prepared by Jeong and coworkers. They proved non-canonical functions of AIMPs can be used as indicators for the degeneration process of peripheral nerve via a series of experiments. The data is well-organized and convincing. I recommend the acceptance of this paper on Materials.
Author Response
Point 1: Overall a good paper prepared by Jeong and coworkers. They proved non-canonical functions of AIMPs can be used as indicators for the degeneration process of peripheral nerve via a series of experiments. The data is well-organized and convincing. I recommend the acceptance of this paper on Materials.
Response 1: We thank you for your comment and your reviewer’s works.

Reviewer 3 Report
The study follows expression changes of the proteins AIMP1, AIMP2, and AIMP3 throughout nerve injury and subsequent recovery, showing a distinct pattern of expression regulation. While the data presented is clean and well presented, the following comments should be addressed:
- to make the fluctuation of AIMP1-3 in Figure 1 and 2 more convincing and to help come to the conclusion of a noncanonical effect of AIMP, the authors should add more controls beside GAPDH and beta-actin. Do other proteins involved in the formation of the multisynthetase complex show the same pattern? Is this change due to an increase in general translation or specific for AIMPs? The lack of response by GAPDH and beta actin could be due to longer protein-halflife. The discussion later on focuses on a non-canonical function of AIMP1-3 but the data does not provide much information on why.
- Figure 5 is not very informative - I appreciate that the point is to show that nothing changes but is this due to the high initial levels of AIMPs in untreated cells? How do they compare with Schwann cells, both during injury and control? Legend for d) is missing.
- In order to state that AIMPs are essential for trans-dedifferentation, AIMP levels should be manipulated (ideally by a knock down or knock out). Correlation alone, especially without the adequate controls, is not sufficient for such a statement. In general, the discussion is rather speculative... Without this (over-)interpretation the manuscript would still be an interesting note on the fluctuation of AIMP levels.
- I am puzzled by the choice of the authors to submit to "Materials" - there seem to be no overlap between the study presented and the scope of the journal (and previously published work).
Author Response
Point 1: to make the fluctuation of AIMP1-3 in Figure 1 and 2 more convincing and to help come to the conclusion of a noncanonical effect of AIMP, the authors should add more controls beside GAPDH and beta-actin. Do other proteins involved in the formation of the multisynthetase complex show the same pattern? Is this change due to an increase in general translation or specific for AIMPs? The lack of response by GAPDH and beta actin could be due to longer protein-halflife. The discussion later on focuses on a non-canonical function of AIMP1-3 but the data does not provide much information on why.
Response 1: We thank you for your comment and agree with your concern regarding further evaluation of AIMP expression patterns using other controls. However, qPCR and Western blot analyses were additionally performed to support our in vivo AIMPs imaging using fluorescence-based analysis showing same expression patterns of AIMPs compared with the quantitative data. Thus, our quantification of mRNA and protein expression of AIMPs in Figure 1 and 2 could be supported by Figure 3, 4 and 6 and could not be necessary to perform additional quantification.
Point 2: Figure 5 is not very informative - I appreciate that the point is to show that nothing changes but is this due to the high initial levels of AIMPs in untreated cells? How do they compare with Schwann cells, both during injury and control? Legend for d) is missing.
Response 2: We thank you for your comment and agree with your concern regarding no alteration of AIMPs expression in Figure 5. Instead of Schwann cells, satellite cells take the place of a role for supporting DRG neuronal cell bodies in DRG regions. In Figure 5, we intended to show cell-specific functions of AIMPs in Schwann cells, but not neurons or satellite cells. Additionally, following your comment, we have added “Western blot analysis showed expression patterns of AIMPs in dorsal root ganglia (DRG) with or without nerve injury. β-actin was used as loading control” to legend of Figure 5c in revised manuscript. The revised portion of the manuscript and new text appear on page 7, lines 200 ~ 202.
Point 3: In order to state that AIMPs are essential for trans-dedifferentation, AIMP levels should be manipulated (ideally by a knock down or knock out). Correlation alone, especially without the adequate controls, is not sufficient for such a statement. In general, the discussion is rather speculative... Without this (over-)interpretation the manuscript would still be an interesting note on the fluctuation of AIMP levels.
Response 3: We thank you for your comment and agree with your concern regarding further evaluation for characterizing AIMPs. However, in in vivo experiments, it is impossible technically to figure out the characterization of AIMPs because effective inhibitors of AIMPs have not found. Thus, we focused on for the first time demonstrating the cell-specific morphological characteristics of AIMPs upregulation irrelevant to their canonical protein synthetic function.
Point 4: I am puzzled by the choice of the authors to submit to "Materials" - there seem to be no overlap between the study presented and the scope of the journal (and previously published work).
Response 4: We thank you for your comment. This special issue is “Fluorescent Probes for Imaging and Detection” and our main technique “immunofluroscence detection” to obtain fine in vivo imaging is usually used with fluorescent macromolecular materials. Thus, we think that our results include in the scope of this special issue.

Round 2
Reviewer 3 Report
Thank you for clarifying point 4. I agree that point 3 is beyond the scope of this study and would need a lot of additional work.
Regarding Point 1 and 2 I don't think my questions were properly addressed.
The easiest way to answer point 1 is to show a Western blot of other proteins in the translation apparatus, especially tRNA synthetases, over the time course of the model to provide some evidence that it is a noncanonical function.
For point 2, the easiest way to answer would be to compare levels of AIMP1 between the different cells types.
Alternatively, if no additional experiments are planned, the authors should weaken the statement in the last sentence, as there is insufficient data to support it:
"Taken together, these observations suggest that the non-canonical
functions of AIMPs are essential for the trans-dedifferentiation of Schwann cells in dysfunctional peripheral nerves."
Neither the contribution of noncanonical functions alone nor AIMPs being essential has been shown by the experiments. I think that even without this claim it is an interesting, clean study on AIMP fluctuation and worth publishing.
Author Response
Point 1: to Regarding Point 1 and 2 I don't think my questions were properly addressed. The easiest way to answer point 1 is to show a Western blot of other proteins in the translation apparatus, especially tRNA synthetases, over the time course of the model to provide some evidence that it is a noncanonical function. For point 2, the easiest way to answer would be to compare levels of AIMP1 between the different cells types. Alternatively, if no additional experiments are planned, the authors should weaken the statement in the last sentence, as there is insufficient data to support it: "Taken together, these observations suggest that the non-canonical functions of AIMPs are essential for the trans-dedifferentiation of Schwann cells in dysfunctional peripheral nerves." Neither the contribution of noncanonical functions alone nor AIMPs being essential has been shown by the experiments. I think that even without this claim it is an interesting, clean study on AIMP fluctuation and worth publishing.
Response 1: We thank you for your comment and agree with your concern regarding further evaluation of AIMP expression patterns using other controls. Following your comments. We toned down our suggestion in Section 4.2 “Taken together, these observations suggest that functions of AIMPs in dysfunctional peripheral nerves may involve in trans-dedifferentiation of Schwann cells. However, further evaluation is needed to strengthen the non-canonical functions of AIMPs on the trans-dedifferentiation of Schwann cells.” The revised portion of the manuscript and new text appear on page 10, lines 282 ~ 285.
